# The Drunkard's Odometry:
# Estimating Camera Motion in Deforming Scenes

**David Recasens**
University of Zaragoza

**Martin R. Oswald**
ETH Zurich, University of Amsterdam

**Marc Pollefeys**
ETH Zurich, Microsoft

**Javier Civera**
University of Zaragoza

## Abstract

Estimating camera motion in deformable scenes poses a complex and open research challenge. Most existing non-rigid structure from motion techniques assume to observe also static scene parts besides deforming scene parts in order to establish an anchoring reference. However, this assumption does not hold true in certain relevant application cases such as endoscopies. Deformable odometry and SLAM pipelines, which tackle the most challenging scenario of exploratory trajectories, suffer from a lack of robustness and proper quantitative evaluation methodologies. To tackle this issue with a common benchmark, we introduce the Drunkard's Dataset, a challenging collection of synthetic data targeting visual navigation and reconstruction in deformable environments. This dataset is the first large set of exploratory camera trajectories with ground truth inside 3D scenes where every surface exhibits non-rigid deformations over time. Simulations in realistic 3D buildings lets us obtain a vast amount of data and ground truth labels, including camera poses, RGB images and depth, optical flow and normal maps at high resolution and quality. We further present a novel deformable odometry method, dubbed the Drunkard's Odometry, which decomposes optical flow estimates into rigid-body camera motion and non-rigid scene deformations. In order to validate our data, our work contains an evaluation of several baselines as well as a novel tracking error metric which does not require ground truth data. Dataset and code: `https://davidrecasens.github.io/TheDrunkard'sOdometry/`

## 1   Introduction

Deformable scenes are among the most challenging cases for visual navigation and multi-view reconstruction. They may also be among the ones with the most relevant potential applications, ranging from the reconstruction of deforming objects [1], faces [2], hands [3], human bodies [4] (or animals' [5]), clothing [6]) or the body interior for medical applications [7, 8, 9, 10]. Among all potential applications for mapping and navigation in non-rigid scenes, medical ones stand out as very different from the rest and are the target of our work. In certain medical procedures, such as endoscopies, a camera navigates inside the human body, performing *exploratory trajectories* that extend far beyond its field of view. For the rest of the applications mentioned, the camera remains nearly or fully stationary and most views have a high degree of overlap. The field of Non-Rigid Structure from Motion (NRSfM) has experienced significant progress in the last decades, e.g. [11, 12, 13, 14, 15, 16, 17, 18, 19, 20, 21, 22, 23, 24]. However, most of them address small-scale reconstructions and are of limited use in medical applications. For the few exceptions that cover SLAM [25] in deformable scenes (e.g., [7, 8, 10, 26]), there are no clear benchmarks nor datasets to support and track progress in the field.

37th Conference on Neural Information Processing Systems (NeurIPS 2023) Track on Datasets and Benchmarks.

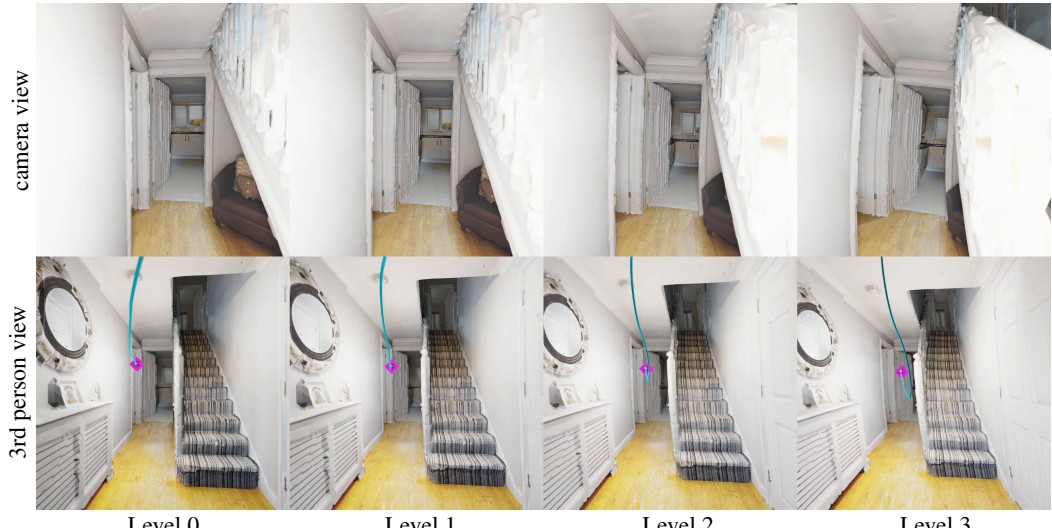

Figure 1: **Sample scene of the Drunkard's Dataset.** The dataset provides various levels of scene deformation. **Top row:** Sample frames from Scene 0 over all difficulty levels $0 - 3$. **Bottom row:** External views showing the ground truth camera trajectory in green and the camera frame in purple. With increasing deformation level the camera motion is more abrupt. See animations with *Adobe Reader*, *KDE Okular* or YouTube.

Our main contribution in this work is motivated by this need for benchmarking exploratory camera motion estimations in deforming scenes. Publicly available datasets imaging deforming scenes do not cover exploratory trajectories, and the ones that do cover them do not have ground truth geometric annotations. Our proposal is a synthetic dataset that we denote as the "Drunkard's Dataset", containing a set of high-resolution RGB images, ground truth depth, optical flow, normal maps and camera pose trajectories in synthetic but texture-realistic deforming scenes. In order to generate a sufficient amount and variety of data, we imported the real-world scanned indoor 3D models of the Habitat-Matterport 3D dataset [27], added dynamic deformations, and generated trajectories within them. Figure 1 shows several data samples with the camera trajectory that advances over time. As the original 3D models are real-world scans, camera trajectory and depth are in consistent metric scale along scenes. To make it a scalable benchmark dataset, every scene was recorded under four different levels of reconstruction difficulty, increasing the deformation and camera trajectory perturbations (observe them in Figure 1). The Drunkard's Dataset is unique in its kind, providing large-scale data in deformable scenes, which will enable both, benchmarking non-rigid navigation and reconstruction methods, as well as sufficient data to evaluate the potential benefits of deep learning. Please visit our project website for further details and access to the dataset and source code.

Scientific progress is in many occasions based on well-established and public benchmarks. This has been the specific case in computer vision research in the last decades, up to the point that the existence of some benchmarks is highly correlated with scientific progress in the field. Data repositories are also essential nowadays not only to benchmark different methods, but also to train deep neural networks. However, collecting large amounts of data with non-rigid content is challenging due to difficulties in annotation, as argued by Li et al. [2]. Indeed, some very popular datasets are synthetic [28, 29].

Capturing large-scale data in sufficiently large non-rigid spaces, in order to benchmark odometry/SLAM methods, is even more challenging. Having ground truth annotations, in particular in the medical domain, hugely increases the challenge. The most popular medical datasets [30, 31, 32, 33] are small, lack geometric ground truth, or both. As a result, comparisons between methods are very often inaccurate, inconclusive or questionable. This motivates our work.

The second contribution of this paper is the Drunkard's Odometry (see Figure 2), a flow-based odometry method for camera motion estimation from RGB-D sequences in deforming scenes. Our method is inspired by the pose estimation of DROID-SLAM [34] and by the 3D scene flow prediction of RAFT-3D [35]. Our novelty compared to both is that our architecture models potential scene deformations.

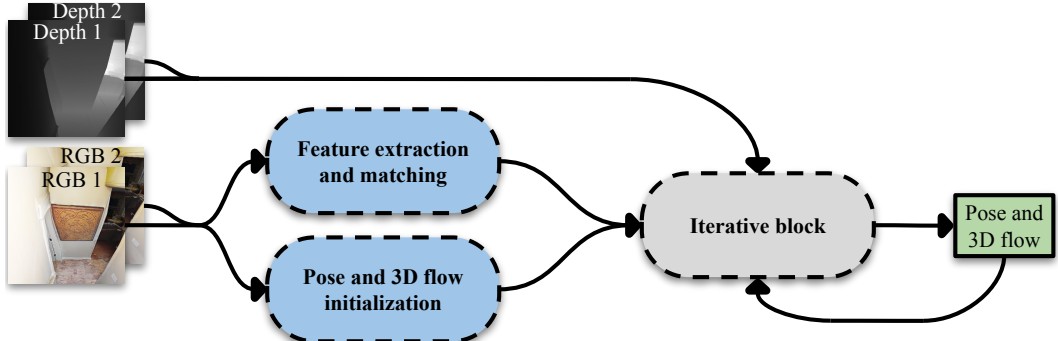

Figure 2: **Overview of the Drunkard's Odometry pipeline.**

As third contribution, in order to validate methods also in non-annotated data, we present a novel metric, the Absolute Palindrome Trajectory Error (APTE). Our novel metric is based on running odometry methods forwards and backwards in a sequence of images, and comparing the pose errors between the first and last frames through different loop lengths. This metric may be useful for validating methods in realistic setups, but it is limited to relative errors between two poses, and hence it is not as informative as metrics using ground truth labels. For this reason, we believe that APTE in real sequences should not be trusted on its own, but as a complement to a larger and more informative set of metrics in our simulated Drunkard's Dataset.

## 2   Related Work

Structure from Motion, visual odometry and visual SLAM methods for rigid scenes are commonly evaluated in a well established set of publicly available datasets [42, 43, 44, 45, 46, 47, 48], with ground truth geometric annotations and under the same metrics, which facilitates comparisons between them and progress in the field. Large-scale annotations for geometric ground truth are quite challenging in real scenarios, and still contain small errors due to the accuracy of the equipment (GPS

| Dataset | Sim/Real | GT | Explo-ratory | # Frames | Domain |
|---|---|---|---|---|---|
| De Aguiar *et al.* [36] | Real | ✓ | ✗ | ∼ 6K | Human bodies |
| FAUST [37] | Real | ✓ | ✗ | ∼ 9K | Human bodies |
| NRSfM Challenge [32] | Real | ✓ | ✗ | - | Objects |
| DeformingThings4D [38] | Sim | ✓ | ✗ | ∼ 122K | Humans & animals |
| ToFu [2] | Sim | ✓ | ✗ | ∼ 20K | Faces |
| EndoMapper [33] | Real | ✗ | ✓ | ∼ 4M | Colonoscopies |
| Hamlyn [39, 30, 40, 41] | Real | ✗ | ✓ | ∼ 93K | Laparoscopies |
| **Drunkard's (ours)** | Sim | ✓ | ✓ | ∼ 416K | Indoor scenes |

Table 1: **Overview of existing non-rigid dataset specifications** compared to our Drunkard's dataset.

or motion capture systems) used. An accurate approximation of the depth and camera pose could be recorded with sensors, but could not obtain such a good quality of optical flow and normal maps. In real-world applications that involve significant deformations, such as endoscopies, these sensors cannot be even equipped. While we could attempt to simulate a small-scale scenario using moving blankets or bouncy castles, it would fall short in terms of capturing a wide range of textures, deformations, and camera trajectories. Virtual environments, on the other hand, offer a superior solution, providing a rich and diverse representation of the complex conditions that are found in real-world settings. Synthetic datasets solve these issues at the price of the sim-to-real gap, and are common for benchmarking methods on navigation and reconstruction on rigid scenes [49, 50, 51, 52, 27]. For similar reasons, synthetic datasets are widely used in other computer vision tasks, such as stereo and flow [53, 28, 54, 29, 55, 56], depth [57, 58], object recognition and segmentation [59], object pose estimation and tracking [60], and scene segmentation, understanding and reasoning [61, 62, 63]. For additional insights and references on the use of synthetic data in deep learning, the reader is referred to the excellent survey by Nikolenko [64].

Note that, in most of the datasets cited, in particular those for faces and humans bodies, the camera is almost stationary. For the few datasets where the camera moves sufficiently, exploring new areas, the covered region is in any case reduced, deformations are small or there is absence of geometric

ground truth (see Table 1). This poses difficulties for benchmarking methods for odometry and SLAM targeting non-rigid environments, such as [7, 9]. Odometry and SLAM methods for deformable scenes are scarce in the literature, being [65, 7, 8, 9, 26, 10, 66, 67] the most representative ones. However, they are all based on feature matching or direct tracking, which make them unstable in challenging sequences. Differently, our Drunkard's Odometry is based on scene flow, which makes it significantly more robust.

# 3 The Drunkard's Dataset

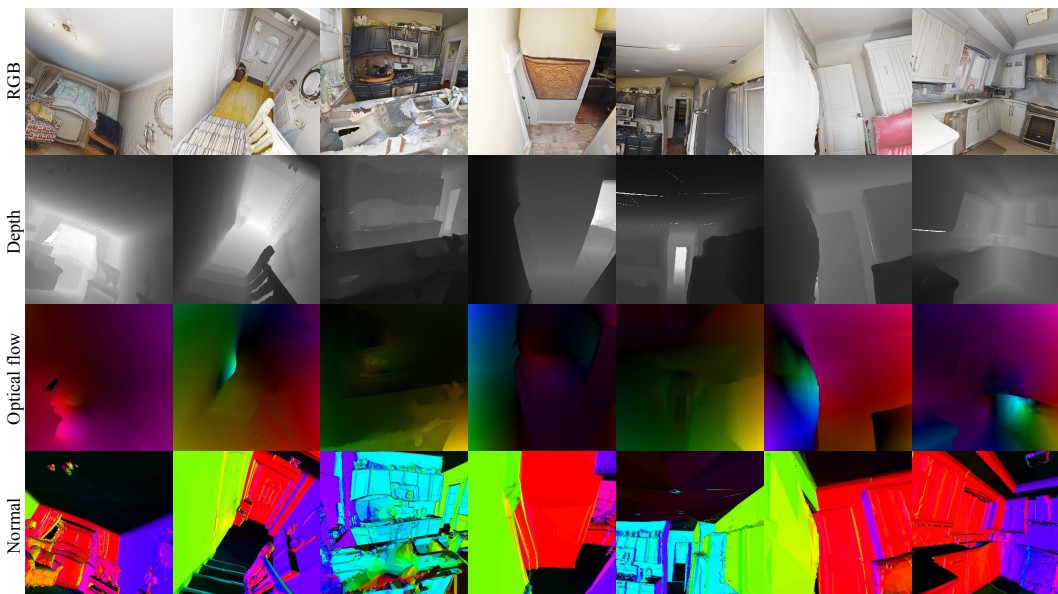

Figure 3: **Sample images of the Drunkard's Dataset** with their corresponding depth, optical flow and normal maps. Non-rigid deformations are simulated by smooth deformations of all scene parts.

The Drunkard's Dataset is a publicly available set of 19 different camera trajectory recordings of 19 different deforming indoor scenes, where each one has been recorded four times, one for each difficulty level. For each level, we generated over 100K frames and recorded camera poses, RGB images, depths, optical flow and normal maps at $1920 \times 1920$, being the camera poses and the depth in consistent real-world metric scale throughout different scenes. Find sample images in Figure 3.

We used Blender [68] to render the deformations of the 19 real-world scanned indoor 3D models of the Habitat-Matterport 3D dataset [27]. We manually designed camera trajectories such that every room in each building is visited once and the camera trajectory returns at the starting point, except for scenes 4, 9 and 14, in which the camera traverses the building three times, but in each loop visiting the rooms in different order. The Blender files are publicly available, along with the scripts we used for generating them, so that anyone can render in-house versions of the Drunkard's Dataset scenes modifying parameters of deformations, pose trajectory, resolution or camera type among others.

As Table 2 details, level 0 stands for zero deformation and camera noise, resulting in a rigid scene and a smooth camera motion, well suited for rigid SfM/SLAM methods. The following levels have an increasing degree of deformation and trajectory noise. Having four levels of difficulty allows to benchmark both rigid and non-rigid methods in a graduated manner.

| Difficulty level | Deformations | Trajectory noise |
|---|---|---|
| Level 0 | 0 | 0 |
| Level 1 | Low | Low |
| Level 2 | Medium | Medium |
| **Level 3** | **High** | **High** |

Table 2: **Overview of the four difficulty levels.** Deformation and camera trajectory noises increases with higher levels. At level 0 there is neither deformation nor camera noise, i.e. it represents a rigid scene and a smooth motion.

## 4 The Drunkard's Odometry

Given a pair of RGB-D images $\{\mathbf{I}_i, \bar{\mathbf{Z}}_i\}$, $\mathbf{I}_i \in \mathbb{R}^{w \times h \times 3}$ and $\bar{\mathbf{Z}}_i \in \mathbb{R}^{w \times h}$, $i \in \{1, 2\}$, our Drunkard's Odometry estimates a dense scene flow $\mathbf{T}^k \in \mathrm{SE}(3)^{w \times h}$ between them, and the relative camera motion $\mathbf{T}_c^k \in \mathrm{SE}(3)$ iteratively ($k \in \{1, \ldots, M\}$ stands for the iterative block step). $\mathbf{T}^k$ contains pixel-wise rigid-body transformations that ideally, i.e. in absence of noise, maps every 3D point $\mathbf{P}_j = \pi^{-1}(\mathbf{u}_j, \bar{z}_j)$ –corresponding to pixel $j \in \mathbf{I}_1$, back-projected from the pixel with image coordinates $\mathbf{u}_j \in \Omega_1$ ($\Omega_1$ is the image domain for $\mathbf{I}_1$) and its sensor (ground truth) depth $\bar{z}_j \equiv \bar{\mathbf{Z}}_1[\mathbf{u}_j]$– to its ground truth equivalent 3D point $\bar{\mathbf{P}}_{j'} = \pi^{-1}(\bar{\mathbf{u}}_{j'}, \bar{z}_{j'})$ back-projected from the true associated pixel with image coordinates $\bar{\mathbf{u}}_{j'} \in \Omega_2$ ($\Omega_2$ as the image domain for $\mathbf{I}_2$) and sensor depth $\bar{z}_{j'} \equiv \bar{\mathbf{Z}}_2[\bar{\mathbf{u}}_{j'}]$. $\pi^{-1}(\mathbf{u}, z)$ stands for the inverse projection model. As the estimated scene flow $\mathbf{T}^k$ might be affected by noise, we will denote as $\mathbf{P}_{j'}^k = \mathbf{T}^k[\mathbf{u}_j]\mathbf{P}_j$ the transformation of $\mathbf{P}_j$ to the local frame of $\mathbf{I}_2$, that in general will not coincide the true corresponding point $\bar{\mathbf{P}}_{j'}$. $\mathbf{T}^k[\mathbf{u}_j]$ is composed of the rigid transformation coming from the camera motion $\mathbf{T}_c^k$ and the one coming from the non-rigid surface deformations $\mathbf{T}_d^k[\mathbf{u}_j] \in \mathrm{SE}(3)$, so that $\mathbf{T}^k[\mathbf{u}_j] = \mathbf{T}_c^k \mathbf{T}_d^k[\mathbf{u}_j]$. If the scene is rigid $\mathbf{P}_{j'}^k = \mathbf{T}_c^k \mathbf{P}_j^k$, and if it is deforming $\mathbf{P}_{j'}^k = \mathbf{T}_c^k \mathbf{T}_d^k[\mathbf{u}_j] \mathbf{P}_j^k$.

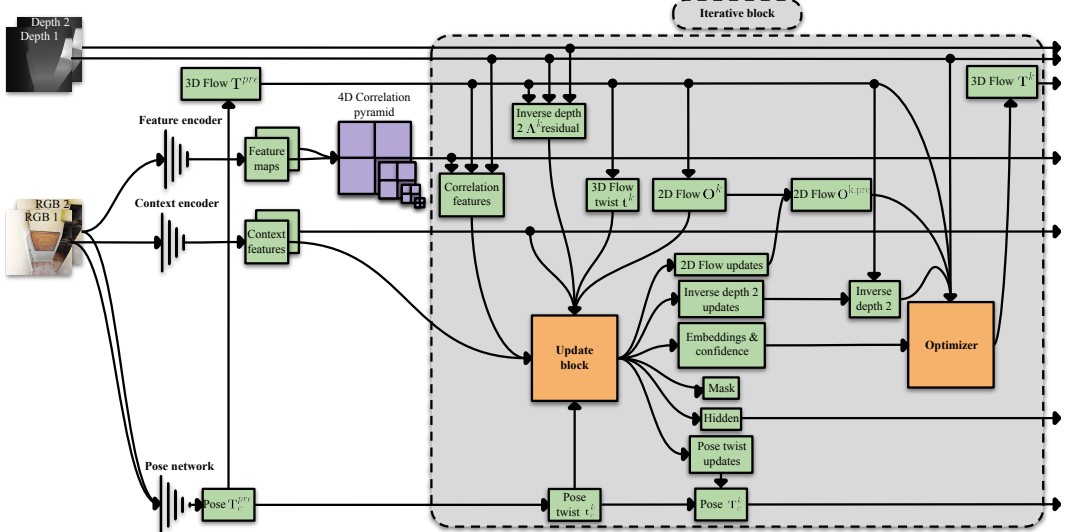

Figure 4: **Drunkard's Odometry architecture in detail.**

Our iterative pose estimation model, shown in detail in Figure 4, is based on the foundations of the 3D flow estimation architecture of RAFT-3D [35], which does not have pose estimation capabilities at all. Firstly, a pose regression network encodes both color images $\mathbf{I}_1$ and $\mathbf{I}_2$ and outputs a initial pre-estimate for the camera motion $\mathbf{T}_c^{\mathrm{pre}}$. This rigid transformation is used for initialization, so that for any $\mathbf{u}_j$ pixel of camera 1 $\mathbf{T}^{\mathrm{pre}}[\mathbf{u}_j] = \mathbf{T}_c^{\mathrm{pre}}$. Additionally, we encode both images $\mathbf{I}_1$ and $\mathbf{I}_2$ in two feature maps, which we use to build a 4D correlation pyramid between the features of all pixel pairs at four different scales, each scale halving the resolution of the previous one. From $\mathbf{T}^k$ and $\bar{\mathbf{Z}}_1$ at each iteration $k$, we can obtain dense 2D pixel correspondences $\mathbf{u}_j \rightarrow \mathbf{u}_{j'}$

$$\mathbf{u}_{j'}^k = \pi\left(\pi^{-1}(\mathbf{u}_j, \bar{z}_j), \mathbf{T}^k[\mathbf{u}_j]\right) , \tag{1}$$

where $\pi(\mathbf{P}, \mathbf{T}[\mathbf{u}])$ stands for the pinhole camera projection model. At the beginning of each iteration, these correspondences are used to sample the correlation features from the fixed 4D correlation pyramid and it is one of the inputs of the update block. The estimated optical flow $\mathbf{O}^k \in \mathbb{R}^{w \times h \times 2}$ coming from these correspondences is also an input for the update block, being $\mathbf{O}^k[\mathbf{u}_j] = \mathbf{u}_{j'}^k - \mathbf{u}_j$.

The update block also takes as input an inverse depth residual obtained from the difference between the estimated inverse depth map $\mathbf{\Lambda}_2^k \in \mathbb{R}^{w \times h}$ and the sensor inverse depth $\bar{\mathbf{\Lambda}}_2 \in \mathbb{R}^{w \times h}$, $\bar{\mathbf{\Lambda}}_2[\mathbf{u}_{j'}^k] = 1/\bar{z}_2[\mathbf{u}_{j'}^k]$, $\forall \mathbf{u}_{j'}^k \in \Omega_2$. Values for $\mathbf{\Lambda}_2^k$ are interpolated from the grid defined by $\lambda_{2,j'}^k = \mathbf{\Lambda}_2^k[\mathbf{u}_{j'}^k] = \mathbf{e}_3^\top \mathbf{P}_{j'}^k$. $\mathbf{e}_3 = (0\ 0\ 1)^\top$ acts as a selector of the third dimension of the 3D point.

A pair of context features that extracts semantic and contextual information from image 1 are also pre-rendered before entering the iterative process using a context encoder network and given to the update block. One context feature map is kept fixed during all iterations and another is used as initialization of the hidden state of the General Recurrent Unit (GRU) [69], which is at the heart of the update block (see the network fine-grained details in the supplementary material).

At each iteration $k$, the estimates for $\mathbf{T}^k$ and $\mathbf{T}_c^k$ are mapped to the Lie algebra with a logarithm map to result in a twist field $\mathbf{t}^k = \log_{\text{SE}(3)}(\mathbf{T}^k)$ and $\mathbf{t}_c^k = \log_{\text{SE}(3)}(\mathbf{T}_c^k)$ before being given to the update block. The update operator outputs a set of updates for the optical flow $\mathbf{O}^k$, for the twist camera pose $\mathbf{t}_c^k$ and for the inverse depth $\mathbf{\Lambda}_2^k$, the hidden state of the recurrent network, a mask and a set of rigid-motion embeddings and confidence maps. These last two maps, next to the updated estimate of $\mathbf{\Lambda}_2^k$ and $\mathbf{O}^k$, and $\mathbf{Z}_1$ are used by the least-squares optimizer block to update $\mathbf{T}^k$ (see details in [35] for this update). Scene flow is updated as $\mathbf{T}^{k+1} = \mathbf{T}^k \exp_{\text{SE}(3)}(\mathbf{t}^k)$.

Internally the network works at $1/8^{th}$ of the original resolution, and the estimated mask at each iteration of the update block is used to perform a convex upsampling to the original resolution of $\mathbf{T}^k$ and intermediate updated 2D flow $\mathbf{O}^{k,\text{pre}} \in \mathbb{R}^{w \times h \times 2}$ by the update block.

The supervision comes from comparing the estimated optical flow $\mathbf{O}^k$ obtained from $\mathbf{T}^k$ and $\bar{\mathbf{Z}}_1$ with Eq. 1 and the intermediate pre-estimated $\mathbf{O}^{k,\text{pre}}$ with the ground truth optical flow $\bar{\mathbf{O}} \in \mathbb{R}^{w \times h \times 2}$, the inverse depth error between $\mathbf{\Lambda}_2$ and $\bar{\mathbf{\Lambda}}_2$, the relative camera pose error of $\mathbf{T}_c$ and the initial guess pre-estimated by the pose network $\mathbf{T}_c^{\text{pre}}$ against $\bar{\mathbf{T}}_c$. The total loss results in

$$\mathcal{L} = \mathcal{L}_{\text{pose}}^{\text{pre}} + \sum_{k=1}^{M} \gamma^{M-k} \left( \mathcal{L}_{\text{flow}}^k + \mathcal{L}_{\text{depth}}^k + \mathcal{L}_{\text{pose}}^k \right) \quad , \tag{2}$$

with $\mathcal{L}_{\text{flow}}^k = \sum_{\mathbf{u}_j} \left( \|\mathbf{O}^k - \bar{\mathbf{O}}\|_1 + w_1 \|\mathbf{O}^{k,\text{pre}} - \bar{\mathbf{O}}\|_1 \right)$ being the optical flow loss term, $\mathcal{L}_{\text{depth}}^k = w_2 \sum_{\mathbf{u}_j} \|\mathbf{\Lambda}_2^k[\mathbf{u}_j] - \bar{\mathbf{\Lambda}}_2[\mathbf{u}_j]\|_1$ the inverse depth loss term, and $\mathcal{L}_{\text{pose}} = w_3 \|\log_{\text{SE}(3)}\left(\mathbf{T}_c \bar{\mathbf{T}}_c^{-1}\right)\|_1$ and $\mathcal{L}_{\text{pose}}^{\text{pre}} = w_4 \|\log_{\text{SE}(3)}\left(\mathbf{T}_c^{\text{pre}} \bar{\mathbf{T}}_c^{-1}\right)\|_1$ the relative camera pose loss terms. $w_l$ stands for the relative weight of the $l^{th}$ loss term and $\gamma$ weights each loop.

## 5 Experiments

In this section, we show the evaluation results of our Drunkard's Odometry against several relevant baselines in two non-rigid datasets: our synthetic Drunkard's and the real Hamlyn data [39, 30, 40, 41].

**Drunkard's Setup.** We trained our Drunkard's Odometry in all scenes of the Drunkard's Dataset except the test ones, with around $90 - 10\%$ ratio for training and testing, respectively, of the difficulty level 1 with an input resolution of $320 \times 320$, batch size of 12, learning rate of $10^{-4}$, Adam optimizer [70], weight decay of $10^{-5}$, 12 iterations of the iterative block during training and test, hyperparameters $w_1 = 0.2$, $w_2 = 100$, $w_3 = 200$ and $w_4 = 6$, and during 10 epochs ($\sim 3$ days) on a single RTX Nvidia Titan. An ablation study available in the Supplementary Material was performed to obtain the hyperparameter values. Everything was trained from scratch except for the pose network encoder which was pre-trained on ImageNet [71].

**Drunkard's Benchmark.** We put to fight our Drunkard's Odometry in the Drunkard's Dataset benchmark against the gold-standard SfM pipeline COLMAP [72, 73], that searches for matches across all images in an offline manner using only the RGB channels of our images; the robust and accurate DROID-SLAM [34] that uses a combination of local-online and global-offline bundle adjustment refinements, trained also in virtual environments of buildings with optical flow and with a well-proved generalization capability between datasets, evaluated using the same RGB-D images as Drunkard's Odometry; and the frame-to-frame tracking designed for non-rigid endoscopic scenes Endo-Depth-and-Motion (EDaM) [9] that here uses the RGB plus the single-view estimated depth maps by a network trained self-supervisely on monocular images of KITTI [43]. The non-rigid SfM SD-DefSLAM [26] was tested but fails in the beginning of the sequences. Complex non-rigid optimization methods such as this one are unstable and tend to fail in difficult sequences with complicated 3D surfaces and abrupt camera trajectories like the ones in our Drunkard's Dataset.

| Scene | Method | Alignment | Level 0 | | | | Level 1 | | | | Level 2 | | | | Level 3 | | | |
|---|---|---|---|---|---|---|---|---|---|---|---|---|---|---|---|---|---|---|
| | | | frames [%]↑ | RPE [cm]↓ | RPE [°]↓ | ATE [m]↓ | frames [%]↑ | RPE [cm]↓ | RPE [°]↓ | ATE [m]↓ | frames [%]↑ | RPE [cm]↓ | RPE [°]↓ | ATE [m]↓ | frames [%]↑ | RPE [cm]↓ | RPE [°]↓ | ATE [m]↓ |
| 0 | COLMAP [72] | Sim(3) | 42 | 0.36 | 0.10 | 0.11 | 42 | 0.85 | 0.18 | 0.12 | 42 | 1.89 | 0.35 | 0.81 | 25 | 3.64 | 0.51 | 1.14 |
| | DROID-SLAM [34] | SE(3) | 100 | 0.77 | 0.28 | 2.38 | 100 | 1.41 | 0.42 | 1.79 | 100 | 2.53 | 0.69 | 1.26 | 100 | 3.58 | 1.01 | 1.00 |
| | EDaM [9] | Sim(3) | 100 | 1.83 | 1.21 | 1.49 | 100 | 1.82 | 1.37 | 1.83 | 100 | 2.05 | 1.72 | 1.95 | 100 | 2.66 | 2.27 | 2.01 |
| | EDaM w/ GT depth | SE(3) | 100 | 2.05 | 1.13 | 2.01 | 100 | 2.28 | 1.28 | 1.78 | 100 | 2.89 | 1.17 | 2.24 | 100 | 3.57 | 2.15 | 2.42 |
| | Drunkard's Odometry | SE(3) | 100 | 0.34 | 0.10 | 0.67 | 100 | 0.59 | 0.16 | 1.08 | 100 | 1.14 | 0.28 | 1.35 | 100 | 1.82 | 0.48 | 1.74 |
| 4 | COLMAP [72] | Sim(3) | 32 | 0.38 | 0.083 | 0.10 | 32 | 1.2 | 0.15 | 0.24 | 32 | 3.12 | 0.30 | 0.90 | 23 | 8.59 | 0.71 | 2.31 |
| | DROID-SLAM [34] | SE(3) | 0 | - | - | - | 0 | - | - | - | 0 | - | - | - | 0 | - | - | - |
| | EDaM [9] | Sim(3) | 100 | 5.50 | 2.16 | 4.85 | 100 | 5.27 | 2.27 | 4.81 | 100 | 5.39 | 2.56 | 4.79 | 100 | 5.88 | 2.96 | 4.90 |
| | EDaM w/ GT depth | SE(3) | 100 | 6.01 | 2.02 | 7.58 | 100 | 5.99 | 2.14 | 7.26 | 100 | 6.31 | 2.40 | 7.94 | 30 | 6.81 | 2.90 | 2.66 |
| | Drunkard's Odometry | SE(3) | 100 | 0.60 | 0.14 | 1.21 | 100 | 0.83 | 0.18 | 1.39 | 100 | 1.43 | 0.28 | 2.46 | 100 | 2.26 | 0.46 | 4.66 |
| 5 | COLMAP [72] | Sim(3) | 100 | 0.40 | 0.08 | 0.20 | 80 | 1.12 | 0.16 | 0.53 | 100 | 3.58 | 0.356 | 1.38 | 31 | 4.95 | 0.46 | 2.45 |
| | DROID-SLAM [34] | SE(3) | 100 | 0.56 | 0.21 | 1.25 | 100 | 1.52 | 0.39 | 1.56 | 100 | 3.16 | 0.67 | 2.43 | 100 | 4.69 | 1.02 | 2.70 |
| | EDaM [9] | Sim(3) | 100 | 3.05 | 1.98 | 2.82 | 100 | 3.13 | 2.11 | 2.73 | 100 | 3.57 | 2.46 | 2.99 | 100 | 4.12 | 2.98 | 2.86 |
| | EDaM w/ GT depth | SE(3) | 100 | 4.63 | 1.95 | 4.00 | 100 | 4.65 | 2.07 | 2.86 | 100 | 5.21 | 2.43 | 4.19 | 100 | 6.01 | 2.91 | 4.03 |
| | Drunkard's Odometry | SE(3) | 100 | 0.45 | 0.13 | 0.47 | 100 | 0.74 | 0.18 | 0.70 | 100 | 1.44 | 0.29 | 1.24 | 100 | 2.40 | 0.49 | 2.45 |

Table 3: **Trajectory errors for Drunkard's test scenes for all difficulty levels.** Note that COLMAP is an offline method and is only shown for reference. Our odometry method mostly outperforms the compared online odometry methods. Best results are highlighted as **first** , second , and third .

We chose scenes 0, 4 and 5 as the test ones with 3.816, 4.545 and 1.655 frames respectively. The rest of the scenes were used for training. The trajectories estimated by COLMAP and EDaM are compared against the ground truth after Sim(3) alignment, as they are up-to-scale. DROID-SLAM and our Drunkard's are aligned to the ground truth with a SE(3) transformation, as the RGB-D input allows them to estimate the real scale. As using Sim(3) against SE(3) is not totally fair since the former applies a scale correction, we also test EDaM using the ground truth depth maps so we can apply SE(3) alignment. The reported metrics are: Relative Position Error (RPE) for translation and rotation, that measures the local accuracy of the estimated trajectory against the ground truth between consecutive frames, and the Absolute Trajectory Error (ATE) for translation that computes the global consistency between both trajectories (see [74] for details).

For each sequence, the percentage of registered frames over the total is shown, a metric in which COLMAP shows poor performance. As a consequence, its trajectory metrics are influenced positively as it excludes frames that are challenging to track and probably would have increased the error. This is beneficial in particular for the ATE, as it takes into account the global consistency rather than frame-to-frame errors like the RPE, and happens earlier in higher deforming scenes. Also note that DROID-SLAM is very GPU-memory demanding, in part because of the final global bundle adjustment and it is not able work with long sequences like Scene 4. However, our Drunkard's Odometry and EDaM are more robust, partly due to tracking only between adjacent frames.

Table 3 shows our results. Note that our Drunkard's Odometry practically always outperforms DROID-SLAM and EDaM (with and without ground truth depth) in RPE and ATE, also in rigid scenes, even if our model is trained exclusively in non-rigid scenes and does not use loop closure or full bundle adjustment. The gap is significantly larger at higher deformation levels, for which the Drunkard's Odometry errors increase much less. This demonstrates that our method is able to generalize at predicting surface deformations. Only COLMAP is able to outperform our Drunkard's Odometry in ATE. Note, in any case, that COLMAP has a much lower recall, as it only estimates camera motion for a substantially lower percentage of frames. If we focus on RPE, a more fair metric in this case, our Drunkard's Odometry is on par to COLMAP at lower deformation levels, and clearly outperforms it at higher ones.

**Validation in real endoscopies.** We used the Hamlyn dataset [30], that contains intracorporeal endoscopic RGB scenes with weak textures, deformations and reflections. Specifically, we chose scenes 1 and 17 (see Figure 5), which are significant exploratory ones. Most of Hamlyn's videos have very small camera motions, being of no interest for benchmarking odometry methods. We slightly cropped the images to remove black pixels at the borders. Depth data was taken from the public tracking test data of EDaM [9] which was estimated by a single-view dense depth network trained in a self-supervised manner in all Hamlyn scenes except for the test ones. Note that this depth does not have the same quality as the real ground truth one from the Drunkard's Dataset.

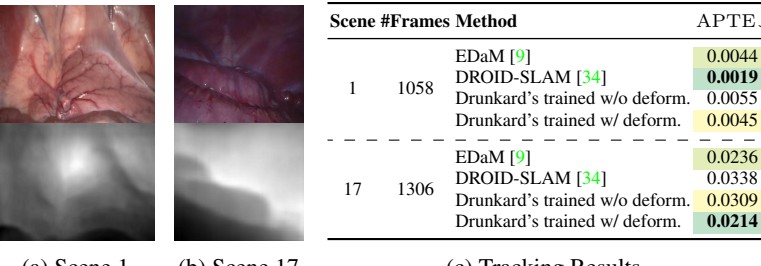

| Scene | #Frames | Method | APTE↓ |
|---|---|---|---|
| 1 | 1058 | EDaM [9] | 0.0044 |
| | | DROID-SLAM [34] | **0.0019** |
| | | Drunkard's trained w/o deform. | 0.0055 |
| | | Drunkard's trained w/ deform. | 0.0045 |
| 17 | 1306 | EDaM [9] | 0.0236 |
| | | DROID-SLAM [34] | 0.0338 |
| | | Drunkard's trained w/o deform. | 0.0309 |
| | | Drunkard's trained w/ deform. | **0.0214** |

(a) Scene 1      (b) Scene 17      (c) Tracking Results

Figure 5: **Sample color and depth frames and** APTE **results for the Hamlyn test videos.**

Since the Hamlyn dataset lacks ground truth camera poses, we propose a novel ground truth-free trajectory metric to measure the quality of the estimated odometries. The key idea is to generate loopy videos by duplicating and reversing a given image sequence and concatenating it to the end of the original sequence. The generated "palindrome video" is twice as long as the original sequence and any tracking method should ideally loop back to its starting position. We then simply measure the discrepancy between start and end pose and accumulate it over various loop lengths. We denote the new metric as the Absolute Palindrome Trajectory Error (APTE) and define it as

$$\text{APTE} = \frac{\sum_{k=1}^{N} \text{APTE}_k}{N} \qquad \text{with} \quad \text{APTE}_k = \left\| \left( \prod_{j=1}^{k} \mathbf{T}_{cj}^{back} \right) \left( \prod_{j=1}^{k} \mathbf{T}_{cj} \right) \right\|_2, \quad (3)$$

where $N$ is the total number of frames and $\text{APTE}_k$ the translation Root Mean Squared Error (RMSE) of between the first and last pose of the sub-trajectory loop $k$. This last pose of the loop $k$ is the result of sequentially applying to the starting camera pose –the identity $\mathcal{I} \in \text{SE}(3)$– the first $k$ estimated relative camera poses $\mathbf{T}_c$ for the scene plus the first $k$ estimated relative camera poses for the same scene but ran backwards $\mathbf{T}_c^{back}$. Then, the $\text{APTE}_k$ is the module of the translation of this last estimated pose. We note that the APTE error is trivially minimized if all transformations are zero or close to the origin, but these cases are easy to check visualizing the trajectory.

Table 5c shows our APTE metric in the Hamlyn test scenes for the baselines that are able to track the whole scene (EDaM and DROID-SLAM) and two versions of our Drunkard's Odometry, one trained only in rigid scenes of the Drunkard's Dataset, i.e. scenes from difficulty level 0, and other in deformable scenes, specifically from level 1. SD-DefSLAM [26] was also tested here and breaks after a few frames, far from the full lenght of the trajectory, even having originally been built and tested in Hamlyn. In consequence, it is not possible to compute the APTE since we need the full trajectory run forward and backward. Before computing the APTE, all the trajectories are scaled with the one given by $\text{Sim}(3)$ alignment, having as reference the trajectory estimated by EDaM, since it showed a good qualitative tracking performance in Endo-Depth-and-Motion [9].

Figure 6 helps to visualize how the $\text{APTE}_k$ values evolve as the length of the loops grows. Notice that a longer loop does not always mean higher $\text{APTE}_k$. This is because there can be some specific problematic frames with a bad pose estimation that induce a drift accumulation in the following ones getting worse $\text{APTE}_k$ measures. However, this can be reversed if the tracking recovers partially during the way back in the loop. The single APTE values shown in the main paper are the average of all the $\text{APTE}_k$ measures.

Figure 7 collects the estimated 3D camera trajectories in the Hamlyn test scenes for all the baselines. It is noteworthy how DROID-SLAM is incorrectly producing sharp and chaotic trajectories around the origin, particularly evident in Scene 1 and partially in Scene 17. This can be further observed in Figure 6b, where the plot of DROID-SLAM becomes similar to the others in the second half of the video. The other methods produce smoother and plausible estimates. The qualitative evaluation of visualizing the trajectories and the quantitative by the APTE complement each other. The former helps in recognizing when a method is producing entirely incorrect camera poses, while the latter provides an objective measure for comparing methods when the trajectories already appear good and plausible.

One clear conclusion we can extract is that our model performs better if it is trained in non-rigid scenes rather than rigid. Again, this shows its capacity to learn deformation patterns and retrieve

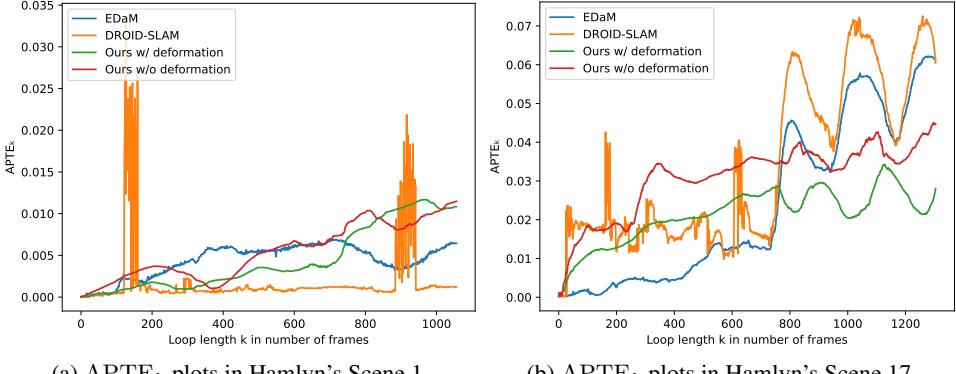

(a) APTE$_k$ plots in Hamlyn's Scene 1.

(b) APTE$_k$ plots in Hamlyn's Scene 17.

Figure 6: **Camera tracking performance on the Hamlyn dataset.** APTE$_k$ values along the $N$ different $k$-frames-length-loops in Hamlyn's scenes 1 (a) and 17 (b) for EDaM [9], DROID-SLAM [34] and Drunkard's Odometry (ours) with and without having been trained in deformable scenes of the Drunkard's Dataset (i.e. trained in level 1 and 0, respectively). The training on deforming scene substantially improves the performance of our method.

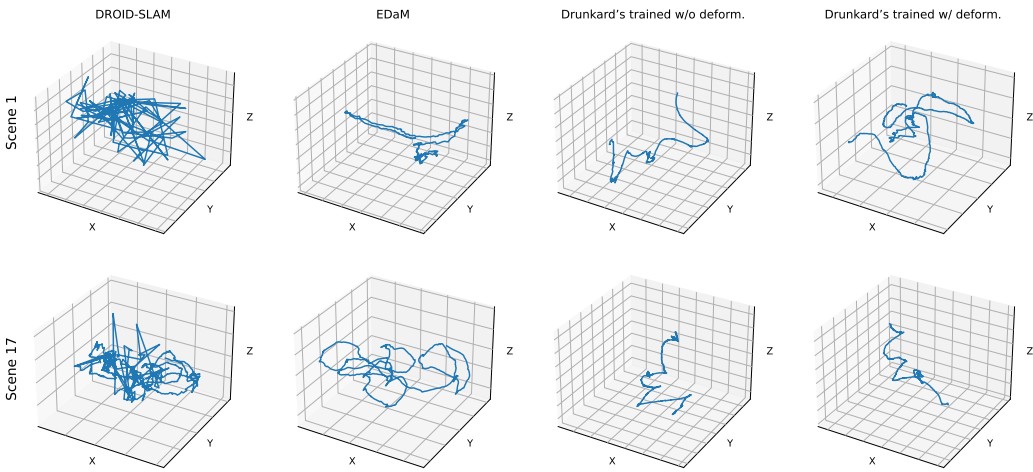

Figure 7: **Estimated 3D trajectories in Hamlyn** test scenes for different methods. See animations with *Adobe Reader*, *KDE Okular* or YouTube.

a more stable camera pose under real-world non-rigid challenging scenes, and even trained in a different domain. Besides, DROID-SLAM's superiority of APTE in Scene 1 results invalidated by observing how it completely fails to estimate the trajectory in Figure 7. This validates the robustness of the Drunkard's Odometry pipeline in addressing the domain gap problem. Another interesting point to comment on is that the advantage of our Drunkard's Odometry over EDaM may not be clear enough. Their performances are similar in Scene 1, while Drunkard's is better for Scene 17. The evaluation in our dataset shows that our Drunkard's Odometry is significantly better than EDaM. This indeed shows the motivation and benefit for our Drunkard's Dataset: looking at Table 3 we can confidently claim that our Drunkard's Odometry is better than EDaM for camera trajectory estimation. The apparent disagreement with the Hamlyn evaluation may arise from the fact that deformations are not graduated in Hamlyn, and qualitatively they are very small, which may benefit rigid methods.

**Limitations.** Our Drunkard's Dataset's most clear limitation is that it is synthetic. However, as we argued, we believe that the difficulties for acquiring high-quality data with ground truth annotations in the target application domains motivates their use. Notice that it is not possible to record true ground truth optical flow in real-world sequences, as we do not have access to the exact pixel motion information. It is precisely this optical flow availability which unlocks the use of powerful flow-based models trained in synthetic deformable data which generalize well to real non-rigid scenes. Despite the real indoor images of the Drunkard's Dataset may resemble real deformable scenes in texture and

shape, such as bouncy castles, moving fabrics or canvas, they are far from the medical application environment. Still, we think that generating medical data with realistic deformations, fluids or textures in large scale is out of reach. As a proof, such data does not exist yet and motivates the use of the Drunkard's data. Our Drunkard's Odometry has all the limitations inherent to a frame-to-frame tracking method. Drift accumulates very quickly, and even if our sequences are loopy we do not either detect loop closures or correct our trajectories based on them. However, the SLAM literature shows that SLAM methods (e.g., [75]) can be built on top of odometry ones (e.g., [76]).

## 6 Conclusions

Estimating camera motion in deformable scenes is a challenging research problem relatively under-explored in the literature, and for which a lack of clear benchmarks slows down research progress. In this work, we created the Drunkard's Dataset, a large-scale simulated dataset with perfect ground truth and a wide variety of scenes and deformation levels to train and validate deep neural models. In addition, we propose the Drunkard's Odometry method for deformable scenes to validate our dataset. The method minimizes a scene flow loss, but as its main contribution, intrinsically decomposes the estimated twist flow into two components: The majority of motion is aimed to be explained by a rigid-body camera motion, and all remaining motion is explained by scene deformations. In contrast to most existing works our method does not require a static scene part for estimating a reference coordinate frame which is crucial in fully deforming scenarios like endoscopy. To also assess odometry estimates in the absence of ground truth data, we further define a novel ground truth-free metric for trajectory evaluation that measures the cyclic consistency of a tracking algorithm. Both the dataset and source code for our baseline method are publicly available. Our experimental results validate our dataset, illustrates its challenges, and also shows that our Drunkard's Odometry is able to outperform relevant baselines.

**Acknowledgements.** This work was supported by the EU Comission (EU-H2020 EndoMapper GA 863146), the Spanish Government (PID2021-127685NB-I00 and TED2021-131150BI00), the Aragon Government (DGA-T45 17R/FSE), and a research grant from FIFA.

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
