# OpenReview forum: "The Drunkard’s Odometry: Estimating Camera Motion in Deforming Scenes"
_NeurIPS.cc/2023/Track/Datasets_and_Benchmarks — NeurIPS 2023 Datasets and Benchmarks Poster_

### Official Review · Reviewer_s3zk · 2023-07-21
**Interesting work with some room for improvements**

**Rating:** 6
**Confidence:** 3
**Correctness:** The dataset looks reasonable and inte…

**Strengths:**

- Interesting dataset that has a decent size with a novel setting.

- The proposed method achieves impressive results, especially it is transferable to medical applications.

- The proposed metric makes sense especially in absense of ground truth trajectory.

**Additional Feedback:**

The second page cannot be viewed properly on my iPad (the whole page is shifted to the left by half of the width), but it works fine on PC. I'm not sure if others have the same issue.

**Clarity:**

I am not particularly satisfied with the explanation of the Drunkard's Odometry. Please see Opportunities For Improvement.

**Documentation:**

I think the implementation details can be better presented, but since the source code is available, I believe reproduction of its performance would not be a significant issue.

**Ethics:**

No ethics issue to my awareness.

**Limitations:**

I think the limitations have been adequately discussed. In particular, I do not think using synthetic data is a limitation but it is defininately ok to include it.

**Opportunities For Improvement:**

- About motivation. I am still unconvinced why would trajectory recontruction would be useful in endoscopies, especially when the trajectory may be far from accurate (>1 m error). However, I suggest looking at odometry in video with fast movement (motion blur), that also introduce substantial deformation of scene due to camera distortion.

- Dataset implementation lacks details.
    - Table 2, it will be much clearer to provide numbers instead of "low / medium / high".
    - How's deformation implemented exactly? i.e. what blender functions are used?

- Presentation of the Drunkard's Odometry needs improvement.
   - It is difficult to associate text with different part of Fig 3 (i.e., line 149, it will be much clearer to know it refers to the 4D correlation pyramid, in fact this part can be illustred individually). I suggest split Fig 3 into a simple overall pipeline chart, and draw other modules in separate figures to illustrate their details.
   - Overall the delivery is very overwhelming (too much symbols that contribute to overall complexity, especially in the absence of clear illustrations).
  - Some symbols are not clearly defined. e.g., What does $\Omega$  in line 135 mean? Does the hat (i.e., $\bar{Z_{i}}$) mean ground truth?
  - Can you highlight how is the method designed to combat deformation?

- Contradictory claims. In line 73-74, it is stated that "APTE ... should not be trusted on its own". I suggest putting this statement in the limitations section instead of where it is now, because such statement undermines the contribution of APTE and leads to incoherency.

- Experiments and analysis.
    - It will be more convincing to also evaluate on visual odometry dataset with rigid scenes other than your own dataset (line 221).
    - In line 237, depth quality is mentioned. I am very curious to see the depth estimation in real endoscopies. Perhaps reconstructing the trajectory with depth and RGB pixels will be a very spectacular visualization.

- About APTE, would adding a distance term in the denominator help eliminate trivial situations of zero transformation? If the distance is very small, the error will be amplified. In other words, the error is reduced if the total distance is large.

- Formating. Large gap under Figure 2.

**Relation To Prior Work:**

I think the coverage is adequate, but other reviewers may have more suggestions.

**Summary And Contributions:**

This works introduces a novel deformable / non-rigid odometry dataset "The Drunkard's Dataset", along with a new flow-based odometry algorithm "The Drunkard's Odometry", and a new metric APTE to evaluate methods in absence of ground truth trajectories.

---

> ### Author Response · Authors · 2023-08-16
> **Part 1**
>
> 1) About motivation. I am still unconvinced why would trajectory recontruction would be useful in endoscopies, especially when the trajectory may be far from accurate (>1 m error). However, I suggest looking at odometry in video with fast movement (motion blur), that also introduce substantial deformation of scene due to camera distortion.
>
> We appreciate the reviewer's perspective and insights. However, it should be noted that the >1m error happens with the Absolute Trajectory Error (ATE) metric (not in the relative metrics between frames), which is dependent on the total length of the trajectory, and in the Drunkard’s Dataset (which has real-world scale) instead of the endoscopic dataset. For example, to realize that they are not excessive errors, the ATE in the Scene 0 of the Drunkard’s Dataset, which has a total trajectory length of 73.38m, for the Drunkard's Odometry ranges from 0.67m to 1.74m when going from level of deformation 0 to 3, i.e. 0.91-2.37% of the traveled distance, and the other online methods are normally worse. In endoscopic scenarios, we do not have a ground truth trajectory to evaluate the ATE, but if it scales accordingly it should be in the range of 1-2 centimeters.
>
> Regarding the usefulness of the trajectory in endoscopies, it is a research topic that is growing rapidly due to its enormous potential. It is a critical part of many endoscopic research approaches, like depth estimation [h], SLAM [i] or scene reconstruction [j]. Even the authors from the University of Zaragoza of this work are part of the European funded project Endomapper [k], whose goal is to develop the fundamentals for real-time localization and mapping inside the human body, in order to enhance the information available for endoscopists. Trajectory estimation is critical in this field and the real-world practicality of this research is very diverse. For example, to bring to endoscopy live augmented reality, useful to show to the surgeon the exact location of a tumor that was detected in a tomography, as immersive education, or to provide navigation instructions to reach the exact location where to perform a biopsy. Also, deformable intracorporeal mapping and localization will become the basis for novel medical procedures that could include robotized autonomous interaction with the live tissue in minimally invasive surgery or automated drug delivery.
>
> About the suggestion of looking at videos with motion blur, that would differ from the topic researched in this paper. Here we presented a dataset with the aim of serving as a multi-purpose benchmark for non-rigid tasks, complementary to real ground truth-free datasets, and we presented an odometry as a practical case of use. Thus, in this work we tested the ability of our method to manage and estimate deformations, so testing its robustness under different motion blur conditions is out of scope, but is an interesting research on its own.
>
> 2) Dataset implementation lacks details.
> - Table 2, it will be much clearer to provide numbers instead of "low / medium / high".
> - How's deformation implemented exactly? i.e. what blender functions are used?
>
> We agree that it is relevant to show the deformation parameters and their numeric values about the dataset creation in Blender. We found that those numbers would be cumbersome in the main paper, thus we decided to collect them in a table in the supplementary material. This was also a concern of reviewer etHK, so a table showing the requested parameters can be found in our reply to her/him.
>
> 3) Presentation of the Drunkard's Odometry needs improvement.
> - It is difficult to associate text with different part of Fig 3 (i.e., line 149, it will be much clearer to know it refers to the 4D correlation pyramid, in fact this part can be illustred individually). I suggest split Fig 3 into a simple overall pipeline chart, and draw other modules in separate figures to illustrate their details.
> - Overall the delivery is very overwhelming (too much symbols that contribute to overall complexity, especially in the absence of clear illustrations).
>
> The reviewer is right, we prepared an additional figure with a much more simplified overview to ease the first reading of the paper, and kept the original overview figure too in the paper for the reader that wants to see all the details. We also revised the text focusing on clarity, and we will keep iterating it until the end of the rebuttal period.
>
> 4) Some symbols are not clearly defined. e.g., What does Ω  in line 135 mean? Does the hat (i.e., Zi¯) mean ground truth?
>
> We clarified both aspects in the paper, thanks for the feedback. Ω_1 is the image domain for image 1. The hat indeed means ground truth, so Zi¯ means the sensor or ground truth depth map.

---

> > ### Author Response · Authors · 2023-08-16
> > **Part 2**
> >
> > 5) Can you highlight how is the method designed to combat deformation?
> >
> > Our Drunkard’s Odometry disentangles the flow produced by the camera motion and the one produced by the scene deformation by training the flow network to predict not only the total scene flow, but also the camera motion. As we detail in Section 4 of the paper, by forcing the network to predict camera motion we can estimate the flow produced by camera motion. Subtracting the camera motion flow to the total flow gives the flow caused by scene deformations.
> >
> > 6) Contradictory claims. In line 73-74, it is stated that "APTE ... should not be trusted on its own". I suggest putting this statement in the limitations section instead of where it is now, because such statement undermines the contribution of APTE and leads to incoherency.
> >
> > We prefer to keep such statements in the introduction, as we believe it is part of the motivation for our dataset. Due to the limitations of the metric APTE in datasets without ground truth, and to the limitations of datasets without ground truth in general, we created our Drunkard’s Dataset in which all metrics can be built on ground truth and evaluated in several sequences with graduated difficulties, leading to very solid conclusions.
> >
> > 7) Experiments and analysis.
> > - It will be more convincing to also evaluate on visual odometry dataset with rigid scenes other than your own dataset (line 221).
> >
> > Level 0 of our data corresponds to rigid scenes. We prefer to evaluate our Drunkard’s Odometry in the Drunkard’s data, as the focus of this NeurIPS Datasets and Benchmarks Track should be on data and not on novel methods. The Drunkard’s Odometry was proposed and evaluated against several baselines only to demonstrate the validity, usability and usefulness of our data. Although our method outperforms all the baselines in our dataset, further research out of the paper scope would be needed to develop/demonstrate a SotA method.
> >
> > 8) In line 237, depth quality is mentioned. I am very curious to see the depth estimation in real endoscopies. Perhaps reconstructing the trajectory with depth and RGB pixels will be a very spectacular visualization.
> >
> > That is definitely a point that could generate curiosity, so we added the depth images next to the Hamlyn color frames. In addition, all the colors and their corresponding depth maps are available in the GitHub repository to check out and use. However, reconstructing the scene in 3D is not as simple as just projecting the pixels to the 3D space using the depth maps since due to the deformation a 3D surface point is not in the same place through the time, spoiling the visualization. Further regularization and optimization techniques should be added, which is a research topic on its own.
> >
> > 9) About APTE, would adding a distance term in the denominator help eliminate trivial situations of zero transformation? If the distance is very small, the error will be amplified. In other words, the error is reduced if the total distance is large.
> >
> > This is a good suggestion. As the reviewer suggests, making APTE dependent on a function of the distance (this is how we understood “distance term” as written by the reviewer) would help in downgrading trivial solutions with small translations. The problem we encountered is, which function of the distance should be better to appropriately balance the error term? Can it be even found a term that is dataset-independent? As the focus of this paper is the Drunkard’s dataset, we preferred to keep the metric simple and check qualitatively per each sequence if the estimated trajectory is small and hence this metric is invalid.
> >
> > 10) Formating. Large gap under Figure 2.
> >
> > Thanks for pointing this out. This problem was addressed in the new version.
> >
> > 11) The second page cannot be viewed properly on my iPad (the whole page is shifted to the left by half of the width), but it works fine on PC. I'm not sure if others have the same issue.
> >
> > Figure 1 is an animation that might not be displayed correctly in all readers. We believe this might cause the problem. Although this might be problematic, we believe that the animation is very illustrative on the data, as it displays very well the camera trajectories and scene deformations. We will try to resolve any issues with other viewers for the final version.

---

> > > ### Author Response · Authors · 2023-08-16
> > > **Part 3**
> > >
> > > [h] Shao, S., Pei, Z., Chen, W., Zhu, W., Wu, X., Sun, D., & Zhang, B. (2022). Self-supervised monocular depth and ego-motion estimation in endoscopy: Appearance flow to the rescue. Medical image analysis, 77, 102338.
> > >
> > > [i] Gómez-Rodríguez, J. J., Lamarca, J., Morlana, J., Tardós, J. D., & Montiel, J. M. (2021, May). SD-DefSLAM: Semi-direct monocular SLAM for deformable and intracorporeal scenes. In 2021 IEEE international conference on robotics and automation (ICRA) (pp. 5170-5177). IEEE.
> > >
> > > [j] Wang, Y., Long, Y., Fan, S. H., & Dou, Q. (2022, September). Neural rendering for stereo 3d reconstruction of deformable tissues in robotic surgery. In International Conference on Medical Image Computing and Computer-Assisted Intervention (pp. 431-441). Cham: Springer Nature Switzerland.
> > >
> > > [k] Endomapper website: https://sites.google.com/unizar.es/endomapper

---

> > > > ### Comment · Reviewer_s3zk · 2023-08-25
> > > > **Thank you for the rebuttal**
> > > >
> > > > I would like to thank the authors for the rebuttal, which clears the majority of my concerns. However, through the author's response, I understand this work is highly dedicated to endoscopic research (with some extendability), instead of a bigger picture of odometry in deformable scenes in general. Nevertheless, I am convinced that this dataset will be a valuable asset to the research community. Hence, I will keep my positive score.

---

> > > > > ### Author Response · Authors · 2023-08-28
> > > > >
> > > > > We greatly appreciate the comprehensive review. We are glad to see that our rebuttal has addressed most of the reviewer's concerns.
> > > > >
> > > > > While our work indeed has a real-experimental focus on endoscopic research, we appreciate the reviewer’s understanding of its potential extensibility to broader applications. We are excited about the value our dataset can bring to the research community and are grateful for the positive score.

---

### Official Review · Reviewer_8ABf · 2023-07-24
**A useful dataset for advancing the deformable scene geometry research.**

**Rating:** 7
**Confidence:** 5
**Correctness:** There is an issue with Page #2 of the…

**Strengths:**

- Motivation: The paper effectively highlights the motivation behind the research, addressing a significant challenge in medical applications like endoscopy due to the lack of large-scale datasets for deforming scenes. This sets a strong foundation for the study's importance and relevance.

- Scale of dataset: The scale of the proposed dataset stands out, covering a wide range of scenes with varying deformities (based on levels) and providing comprehensive annotations due to the synthetic nature, including camera poses, RGB images, depth, optical flow, and normal maps. This richness enhances the dataset's value and utility for researchers in the field. Although a good thing of annotations about this dataset is also a point of concern - extensive evaluation (on top of what is already presented) on other tasks would be useful (expanded more in the improvement section below).

- Result on real endoscopy example: The paper demonstrates performance on a real endoscopy example, offering practical insights into its applicability in real-world scenarios. While not perfect, this validation contributes to the credibility of the proposed approach. However, more comments are provided in the next section.

- Reasonable conclusion:  The conclusion effectively emphasizes the model's better performance when trained on deformable scenes compared to rigid scenes, when the evaluation task is a deformable scene (Hamlyn dataset) as demonstrated in Table 3 and Figure 4. This highlights the method's effectiveness and its potential benefits for relevant applications.

Overall, the paper showcases a compelling approach to tackle the camera motion estimation challenge in deformable scenes, providing a diverse and valuable dataset along with a novel data-driven method.

**Additional Feedback:**

Please have a plan for shifting the dataset to a more stable source (e.g. Zenodo or any service which supports versioning), instead of Google Drive.

**Clarity:**

Yes. The paper is well-written and the sections are written in a reasonable manner.

**Documentation:**

Yes, the paper provides sufficient documentation for datasets, including details on data collection, and organization. The dataset submissions include documentation, intended uses, a URL for reviewer access (a GitHub). For benchmarks, there is enough detail to support reproducibility, and the code on Github appears to be well accessible.

A hosting, licensing, maintenance plan is missing. It is not advisable to put the dataset on Google-Drive. Instead, it is recommended to put the dataset on Zenodo for clear versioning.

**Ethics:**

N/A.

**Limitations:**

The authors have adequately addressed the limitations of their work. The checklist provided at the end of the paper is helpful in this regard.

**Opportunities For Improvement:**

- An exciting opportunity for improvement lies in directly comparing the proposed Drunkard's Odometry with hierarchical localization and mapping algorithms [1,2,3] that are also data-driven. Such a comparison could provide a more meaningful assessment of the method's strengths and weaknesses relative to state-of-the-art approaches in the field. It could help establish the method's competitiveness and advantages in scenarios where camera motion estimation approaches excel.

- The comparison presented in Table 3 seems confusing, given that Drunkard's Odometry was trained on certain scenes from the Drunkard's Dataset. It would be beneficial for the authors to address this design decision in the paper. To ensure a fair comparison, readers would be intrigued to see results when both Drunkard's Odometry and other approaches are fine-tuned on the same dataset. This could involve picking any deep learning-based camera estimation method and training it on the Drunkard's Dataset, allowing for a more direct comparison of their performances.

- Clarifying the potential applicability of Drunkard's Odometry to scene flow estimation would be valuable for readers. While it might not be directly comparable to RAFT-3D, an explanation of whether the proposed method can be adapted or used to estimate scene flow independently could open up new possibilities for researchers interested in this specific task.

- Demonstrating the usage of the proposed approach in calibration (bundle adjustment) would be highly beneficial for the community. This would showcase the potential of dynamic deformable moving objects as anchor points for camera calibration, providing valuable insights for researchers working on camera calibration and related tasks.

By addressing these opportunities for improvement, the authors can strengthen the paper's impact, extend the scope of its applications, and offer more comprehensive insights to the research community. The proposed dataset contribution remains commendable, and these suggested enhancements can further solidify the paper's contributions in the field of camera motion estimation in deformable scenes.

[1] Sarlin et al. "From Coarse to Fine: Robust Hierarchical Localization at Large Scale", CVPR 2019
[2] Sarlin et al. "SuperGlue: Learning Feature Matching with Graph Neural Networks", CVPR 2020
[3] Hierarchical-Localization, https://github.com/cvg/Hierarchical-Localization

**Relation To Prior Work:**

Yes.

**Summary And Contributions:**

Summary: The paper addresses the challenging task of estimating camera motion in deformable scenes, where traditional methods rely on static scenes for camera localization. However, this assumption doesn't hold in certain applications like endoscopies. To address this issue, the authors introduce the Drunkard's Dataset, a synthetic collection of data representing varying deformities in scenes. The dataset includes ground truth information for camera poses, RGB images, depth, optical flow, and normal maps, making it a valuable resource for tackling the problem. The authors also propose their novel approach, the Drunkard's Odometry, which leverages the synthetic dataset for camera motion estimation. The paper presents comprehensive results and evaluates various baselines to support their claims.

Contributions:
1. Drunkard's dataset: Synthetic dataset consisting large set of exploratory camera trajectories within deformable scenes. Due to its synthetic nature, precise ground truth annotations are provided. It stands out as one of the first datasets with extensive ground truth information and covers a wide range of deformable scenes with varying levels of deformation (ranging from level 0 with no deformation to level 3 with heavy deformation).
2. Drunkard's Odometry: This method is specifically designed to estimate camera motions from RGB-D sequences in deformable scenes.
3. Ground-truth free metric: Proposal of Absolute Palindrome Trajectory Error (APTE) which is self-consistency based metric. This metric allows for evaluating the performance of camera motion estimation without the need for ground truth data.

---

> ### Author Response · Authors · 2023-08-16
> **Part 1**
>
> An exciting opportunity for improvement lies in directly comparing the proposed Drunkard's Odometry with hierarchical localization and mapping algorithms [1,2,3] that are also data-driven. Such a comparison could provide a more meaningful assessment of the method's strengths and weaknesses relative to state-of-the-art approaches in the field. It could help establish the method's competitiveness and advantages in scenarios where camera motion estimation approaches excel.
>
> We appreciate this improvement idea by the reviewer. We agree our dataset and method can help the research in hierarchical localization and mapping. However, the baseline that the reviewer proposes to run [1,3] retrieves the 6 DoF position of an image with respect to a previous rendered 3D map, that in that paper is being made with Colmap and hence assumes scene rigidity too. The use case for [1,3] is to localize an image that was not used to estimate the 3D map, but in our case the query images would be the same as the ones used by Colmap. Therefore, [1,3] will not give more accurate camera trajectories than the one Colmap already estimates, for which we already demonstrated inferior performance compared to our Drunkard’s. In addition, this task differs substantially from the one in Drunkard’s Odometry and the other evaluated baselines.
>
> Note that [2], although a fantastic paper, refers to 2D feature matching. The focus on our paper and our dataset lies in modeling/learning the 3D deformations of non-rigid scenes and estimating exploratory camera trajectories in such setups.
>
> 2) The comparison presented in Table 3 seems confusing, given that Drunkard's Odometry was trained on certain scenes from the Drunkard's Dataset. It would be beneficial for the authors to address this design decision in the paper. To ensure a fair comparison, readers would be intrigued to see results when both Drunkard's Odometry and other approaches are fine-tuned on the same dataset. This could involve picking any deep learning-based camera estimation method and training it on the Drunkard's Dataset, allowing for a more direct comparison of their performances.
>
> We understand and share the curiosity of the reviewer. DROID-SLAM, like Drunkard’s Odometry, was trained also in virtual environments of buildings with optical flow with a well-proven generalization capability between datasets (this can be found in the original DROID-SLAM paper), and during the evaluation both methods used the same color and ground truth depth maps. If we trained DROID-SLAM in the Drunkard’s Dataset with level of difficulty 1, like our Drunkard’s Odometry, it would converge poorly since it is designed to be trained in rigid scenes and their mathematical foundations would break under total non-rigidity. Therefore, we should train DROID-SLAM in level 0, which would be similar to the data where it is already trained, producing no significant performance difference. In addition, our Drunkard’s Odometry has to be trained with scene deformations so that it can learn them, hence cannot be trained in level 0 only. As the methods should be trained in different data (rigid versus non-rigid) for optimal performance, a comparison between the two would not be fair.
>
> EDaM was evaluated using the depth predicted by a single-view depth network trained self-supervised in the KITTI dataset. Thus, reflecting on the reviewer’s comment, in order to rule out the effect of the training data, we repeated such experiment using the ground truth depth. We added the results in an additional row in Table 3. It can be seen how the metrics are similar and conclusions are the same, reinforcing our belief that fine-tuning in Drunkard’s might not offer a significant advantage, but explicitly modeling deformation as our odometry does, is the critical piece that improves the results.

---

> > ### Author Response · Authors · 2023-08-16
> > **Part 2**
> >
> > 3) Clarifying the potential applicability of Drunkard's Odometry to scene flow estimation would be valuable for readers. While it might not be directly comparable to RAFT-3D, an explanation of whether the proposed method can be adapted or used to estimate scene flow independently could open up new possibilities for researchers interested in this specific task.
> >
> > We acknowledge the identification of this strength by the reviewer despite not having been explicitly highlighted in the paper. It is totally correct that the estimation of the scene flow is not useless but complementary to the predicted camera trajectory in tasks such as scene representation/reconstruction/SLAM/SfM. Having this information can help in the mapping phase. For example, it allows tracking the motion of all the pixels on the screen over time (current active research topic [d, e]), which would add new constraints when optimizing or regularizing the reconstructed scene. Furthermore, this would be useful in rigid and in total or partial non-rigid dynamic environments. As an insight, thanks to having both, scene and camera motion, the scene motion that is not similar to the camera one means it belongs to a dynamic/deformable object, which enables dynamic object segmentation from the static scene, and then it allows use that information for other tasks.
> >
> > 4) Demonstrating the usage of the proposed approach in calibration (bundle adjustment) would be highly beneficial for the community. This would showcase the potential of dynamic deformable moving objects as anchor points for camera calibration, providing valuable insights for researchers working on camera calibration and related tasks.
> >
> > We agree with the reviewer in this comment, demonstrating non-rigid full bundle adjustment or non-rigid SLAM in our Drunkard’s dataset would be interesting, and definitely our data will be relevant for advancing such fields. However, at this moment, both are research lines in early stages and for which there are no clear baselines or open-source code. Note that in our paper we report that we run SD-DefSLAM, a SotA baseline for non-rigid SLAM and it failed catastrophically in our data due to the lack of robustness. We did not find any open-source code for non-rigid SfM in large deformable scenes, as we argue in our paper the research focuses on very small scenes and highly overlapping views.
> >
> > 5) A hosting, licensing, maintenance plan is missing. It is not advisable to put the dataset on Google-Drive. Instead, it is recommended to put the dataset on Zenodo for clear versioning.
> >
> > We agree that having the dataset in a service like Zenodo would have advantages, but we found that these services have relatively low storage limits (50 Gb per dataset in Zenodo). The size of the Drunkard’s data is more than 1,5TB. It is true that for example Zenodo does not put restrictions on the number of datasets, so splitting the data in ~30 Zenodo datasets would be a possibility, but we believe that would make it difficult or inconvenient for download. However, we insist that the data is safe in Google Drive, as we are using our institutional account owned by the University and in addition we count with the funding of the European Union’s Horizon 2020 project Endomapper [f] and we made additional backup copies. Nevertheless, we appreciate the reviewer’s comment and see its motivation, and we already contacted Zenodo in order to explore if it would be possible to increase the size limit.
> >
> > Licensing and maintenance plans were available at the datasheet material in the original submission, in the zip file of the supplementary material. The license (MIT) is also stated in the GitHub repository [g] and we added it to the project page.
> >
> > [d] Harley, A. W., Fang, Z., & Fragkiadaki, K. (2022, October). Particle video revisited: Tracking through occlusions using point trajectories. In European Conference on Computer Vision (pp. 59-75). Cham: Springer Nature Switzerland.
> >
> > [e] Wang, Q., Chang, Y. Y., Cai, R., Li, Z., Hariharan, B., Holynski, A., & Snavely, N. (2023). Tracking Everything Everywhere All at Once. arXiv preprint arXiv:2306.05422.
> >
> > [f] Endomapper website: https://sites.google.com/unizar.es/endomapper
> >
> > [g] GitHub repository: https://github.com/UZ-SLAMLab/DrunkardsOdometry

---

### Official Review · Reviewer_yRQV · 2023-07-25
**a large-scale and challenging benchmark in deformable environments**

**Rating:** 6
**Confidence:** 3
**Correctness:** Yes
**Clarity:** Yes

**Strengths:**

The paper addresses the problem of non-rigid deformations in the camera and the unreliable of current trajectory methods in such situations.  It proposes the first large-scale camera trajectory in deformable environments. The proposed evaluation method is reasonable.
The proposed odometry method achieves promising results on synthetic and real datasets. The paper is well written.

**Additional Feedback:**

no further questions.

**Documentation:**

Yes

**Limitations:**

Yes

**Opportunities For Improvement:**

1. More comprehensive experiments are suggested to demonstrate the necessity of non-rigid deformation learning. For example, on a labeled dataset (such as [1]), compare whether the odometry has the performance gain with pre-training on Drunkard.

2. A fair inference speed comparison is suggested in Table 3.

[1] Y. Li, H. Takehara, T. Taketomi, B. Zheng, and M. Nießner, “4dcomplete: Non-rigid motion estimation
407 beyond the observable surface,” in Proceedings of the IEEE/CVF International Conference on Computer
408 Vision, pp. 12706–12716, 2021.

**Relation To Prior Work:**

Yes. Table 1 shows comprehensive differences between this paper and previous works.

**Summary And Contributions:**

This paper builds an in-door synthetic dataset targeting visual navigation and reconstruction in deformable environments, and a deformable odometry method. The proposed dataset is large-scale and challenging and the idea of decomposing the estimated twist flow into two components is convincing. The proposed odometry method achieves very competitive results on the proposed benchmark and real endoscopies dataset.

---

> ### Author Response · Authors · 2023-08-16
>
> 1) More comprehensive experiments are suggested to demonstrate the necessity of non-rigid deformation learning. For example, on a labeled dataset (such as [1]), compare whether the odometry has the performance gain with pre-training on Drunkard.
>
> We thank the reviewer for this appreciation. However, there does not exist any exploratory non-rigid labeled dataset, apart from our Drunkards’ Dataset. The dataset presented in [1] is not exploratory, it contains videos of non-rigid objects where the camera is static or nearly static, and the method they present does not predict the camera trajectory, but the scene flow. Our work addresses a different problem in which we want to estimate the trajectory and present a dataset of a camera that explores large totally non-rigid scenes that do not fit in the vicinity of the camera's field of view. Current SotA methods that estimate the trajectory in exploratory non-rigid scenes like Def-SLAM [b] and SD-DefSLAM [c] do not use learning, and they present a great lack of robustness in deformable scenes. We mention in the paper that we tried to run SD-DefSLAM in both, Drunkard’s Dataset and Hamlyn, and they fail in the beginning of the Drunkard’s scenes and after a few frames on Hamlyn, even though they were designed to work in that dataset. This demonstrates the research challenges still present in research with non-rigid scenes and motivates the need for our dataset.
>
> 2) A fair inference speed comparison is suggested in Table 3.
>
> We find this a good catch that is interesting for readers to have a better idea of how the different methods perform. However, we think this data fits better in a separate table in the supplementary rather than in Table 3. The Table below was added to the supplementary material showing the approximate inference times per frame for all the evaluated baselines previously. These values come from rounding the total time taken for estimating the full trajectory of Scene 5 of the Drunkard’s Dataset with resolution of 320x320 pixels at level of difficulty 0 divided by the total number of frames (1.655). Our Drunkard’s Odometry is the most efficient of all, in addition to being the most accurate. Note that, even if our Drunkard’s Odometry is based on iterative refined optical flow like DROID-SLAM, in our case we do not perform global Bundle Adjustment, which is what gives us the computational advantage. Further note that Scene 5 is the shortest one among the three tested (0, 4 and 5). For longer sequences, the gap between Drunkard’s Odometry versus DROID-SLAM and Colmap would be higher.
>
> | Method              | Time/frame [ms] |
> |---------------------|-----------------|
> | Colmap              | 1.650           |
> | DROID-SLAM          | 200            |
> | EDaM                | 700            |
> | Drunkard's Odometry | 170            |
>
>
> [b] Lamarca, J., Parashar, S., Bartoli, A., & Montiel, J. M. M. (2020). Defslam: Tracking and mapping of deforming scenes from monocular sequences. IEEE Transactions on robotics, 37(1), 291-303.
>
> [c] Gómez-Rodríguez, J. J., Lamarca, J., Morlana, J., Tardós, J. D., & Montiel, J. M. (2021, May). SD-DefSLAM: Semi-direct monocular SLAM for deformable and intracorporeal scenes. In 2021 IEEE international conference on robotics and automation (ICRA) (pp. 5170-5177). IEEE.

---

### Official Review · Reviewer_etHK · 2023-07-28
**Interesting Problem Formulation and Dataset, but weak practicality**

**Rating:** 6
**Confidence:** 3
**Correctness:** The claims are supported by the appro…

**Strengths:**

- The paper proposed useful synthetic dataset for odometry in deformable scenes, which is quite underexplored problem.
- The dataset is generated in the multiple levels of noise and deformations, which makes dataset be a strong benchmark.
- They proposed robust scene flow-based baseline which works robustly in deformable scene scenarios.
- Quantitative results are well-organized and relevant analyses are given.
- They proposed a novel metric which can be used as supplementary metric where the groudtruth camera poses are inaccessible.

**Additional Feedback:**

- The overview figure is quite complicated and less intuitive. I hope it includes some intermediate outputs as figure, rather than many text boxes.
- The colors in tables to highlight the results are a little confusing. I think using red / blue for first/second may be enough.

**Clarity:**

This paper is clearly written with introducing appropriate mathematical notations.

**Documentation:**

The dataset is well documented.

**Ethics:**

I have no ethics concerns on this paper.

**Limitations:**

The limitations readers may have are well discussed.

**Opportunities For Improvement:**

- I'm not fully convinced by the necessity and practicality of this dataset. I cannot come up with other applications other than endoscopies, which is discussed in the paper, but the domain gap seems too large.
- I'd like to know about how the 3D scene mesh data are deformed (e.g., adding random displacement along normal direction, etc.).
- In figure4c, the gap between EDaM and Drunkard's odometry is very small even in the scene 17. Can you elaborate the advantage of Drunkard's odometry over EDaM?
- The authors claimed that the reason why the performance of Drunkard's for scene1 is worse is that due to the many loops in the scene, which is advantagous for EDaM or Droid-SLAM. Then authors may split the scene into multple clips and evaluate APTE on each clips. If the performance of Drunkards on them is better than other methods, the authors' claims will become more strong, where there is only one number (APTE for scene 17) that supports the authors' claim of effectiveness of their baseline and usefulness for sim-to-real.
- I'm conviced with the usefulness of APTE metric for grountruthless dataset, but is there any visualization method to qualitatively show whether trajectory is well estimated? For example, roughly say, NR-SfM with predicted camera trajectory.

**Relation To Prior Work:**

The relation to prior works is well discussed.

**Summary And Contributions:**

This paper proposes a dataset for estimating camera motion in deformable scenes, which is not usua, but useful for relevant application such as endoscopies. Since it's diffucult to get real world dataset for the task, they generated synthetic dataset by adding dynamic deformations to scanned indoor 3D scenes and manually generating camera trajectories within them. They proposed the baseline architecture, Drunkard's Odometry, and new trajectory metric for dataset without ground truth camera poses like real world endoscopy dataset.

---

> ### Author Response · Authors · 2023-08-16
> **Part 1**
>
> 1) I'm not fully convinced by the necessity and practicality of this dataset. I cannot come up with other applications other than endoscopies, which is discussed in the paper, but the domain gap seems too large.
>
> We thank the reviewer for pointing this out, since it is a common question that comes to mind when one thinks about exploratory motions in real non-rigid scenes. We will elaborate further on the practicality of our dataset here, as this is a critical point of our work.
>
> The domain gap between the Drunkard’s Dataset and real endoscopies is clear, and it was a discussion we had in the initial stages of this work. However, the lack of datasets with exploratory trajectories in non-rigid scenes with ground truth is an ever-present problem in endoscopic 3D reconstruction research. As argued in our paper, the consequence is that most of the research is only evaluated qualitatively or with questionable pseudo-ground truth, which limits the progress of the field. Having ground truth in real endoscopies is almost impossible and trying to minimize the domain gap with endoscopic synthetic data is impractical. There are already synthetic colon datasets, but their size is small and hence deep learning methods cannot be properly evaluated. Further, the domain gap is still present in those datasets, as generating massive data with realistic texture and variability is impractical. Finally, colonoscopy images have other challenges in addition to deformation, such as poor texture, fluids, blur, etc. With all that in mind, we believe that our Drunkard’s Dataset is relevant for the field as it isolates the deformation problem and provides massive data and ground truth that enables solid evaluation of methods that may include deep learning.
>
> Although our initial motivation came from endoscopies, we believe that by isolating non-rigid aspects and having ground truth makes it easier to address this research challenge and makes our data also valuable for other tasks. As additional use cases to the presented non-rigid odometry, our Drunkard’s Dataset is useful for other data-driven approaches such as generic non-rigid reconstruction, non-rigid tracking/registration of object’s shape, motion completion, or deformable point cloud matching. It stands out from the rest of non-rigid datasets as it is the first one exploring large environments and not focusing on small objects that deform captured by a static or near-static camera. Our separation into four difficulty levels makes it versatile to benchmark various non-rigid methods. As real-world applications, our work can also be referred to situations when a robot navigates through a fully deforming environment or its field-of-view does not capture enough static references. Apart from endoscopies, this is the case of,  for example, a camera recording bustling human crowds, swaying clothes, soft sofas, or leafy vegetation moved by the breeze.
>
> We would also note that synthetic datasets, for which there is always a certain domain gap, are common for many computer vision tasks for which annotation is challenging (e.g., Flying Chairs for optical flow or TartainAir for SLAM) or simply as an additional benchmark, having the benefits of perfect ground truth and control over all setup variables (e.g., MultiShapeNet [a] for scene representation).
> We will be happy to continue this discussion if there is any point that is unclear or for which the reviewer is in disagreement.
>
>
> 2) I'd like to know about how the 3D scene mesh data is deformed (e.g., adding random displacement along normal direction, etc.).
>
> Fair comment, that is definitely relevant information about the data creation that we did not reflect in our original submission. We have added it to the supplementary material. Regarding the deformations applied to the 3D meshes in Blender and the perturbations to the camera trajectory, the table below shows the specific values that vary between the four different levels of difficulty. The three mesh deformations were all applied to three different empty planes that are flying randomly throughout the 3D space around the mesh. In consequence, at a certain timestamp, the higher the proximity of a plane to a surface, the higher the amplitude of the rendered deformation. The camera perturbations are implemented as Gaussian noise that affects the rotation and translation of the manually recorded steady camera trajectory. All these mesh and camera parameters increase the amplitude as difficulty grows. All the Blender project files for every scene and level of difficulty are publicly available on the GitHub repository, so anyone can render customized versions of the Drunkard's Dataset and see the full details of the implementation.

---

> > ### Author Response · Authors · 2023-08-16
> > **Part 2**
> >
> > |||||||
> > |---|---|---|---|---|---|
> > | |**Parameter** / **Difficulty level**|0|1|2|3|
> > |||||||
> > |**Mesh**|Cast factor|0|0.01|0.03|0.05|
> > |**Mesh**|Wave height|0|0.05|0.1|0.15|
> > |**Mesh**|Simple deform angle [º]|0|0.5|1|1.5|
> > |**Camera**|Noise strength rotation|0|0.6|1.2|2|
> > |**Camera**|Noise strength translation|0|0.3|0.6|1|
> >
> > 3) In figure4c, the gap between EDaM and Drunkard's odometry is very small even in the scene 17. Can you elaborate the advantage of Drunkard's odometry over EDaM?
> >
> > We understand that with the Hamlyn evaluation, the advantage of our Drunkard’s Odometry over EDaM may not be clear enough. Their performances are similar in Scene 1, while Drunkard’s is better for Scene 17. The evaluation in our dataset shows that our Drunkard's Odometry is significantly better than EDaM. This indeed shows the motivation and benefit for our Drunkard’s Dataset: looking at Table 3 we can solidly claim that our Drunkard’s Odometry is better than EDaM for camera trajectory estimation. The apparent disagreement with the Hamlyn evaluation may come from the fact that deformations are not graduated in Hamlyn and qualitatively they are very small, so they may benefit rigid methods.These comments were added to the last paragraph of the validation in real endoscopies at section 5.
> >
> > In addition, Drunkard’s Odometry inference speed is much faster than EDaM, 170 ms versus 700 ms per frame resolution of 320x320 pixels. A table containing the inference time for all the baselines has been added to the supplementary material and in the reply to reviewer yRQV.
> >
> > 4) The authors claimed that the reason why the performance of Drunkard's for scene1 is worse is that due to the many loops in the scene, which is advantageous for EDaM or Droid-SLAM. Then authors may split the scene into multiple clips and evaluate APTE on each clips. If the performance of Drunkards on them is better than other methods, the authors' claims will become more strong, where there is only one number (APTE for scene 17) that supports the authors' claim of effectiveness of their baseline and usefulness for sim-to-real.
> > 5) I'm convinced with the usefulness of APTE metric for groundtruthless dataset, but is there any visualization method to qualitatively show whether trajectory is well estimated? For example, roughly say, NR-SfM with predicted camera trajectory.
> >
> > These two points are good insights. In the last sentences of the introduction, we already mentioned that the APTE metric was useful to measure the quality of estimated trajectories in ground truth-free datasets, but is not as informative as metrics using ground truth annotations as those in our  Drunkard’s Dataset. There, DROID-SLAM performed clearly worse than Drunkard’s Odometry even in the rigid scenes, which is shocking seeing the superiority of DROID-SLAM in tha Hamlyn’s Scene 1. Motivated by this second point of the reviewer, we decided to visualize in 3D the estimated trajectories of all the baselines in both Hamlyn test scenes. These plots have been added to the experiment section in the paper. With this figure we can see how DROID-SLAM is wrongly producing random poses around the origin, specially remarkable in Scene 1 and partially in Scene 17, as can be contrasted seeing how the plot of DROID-SLAM in the figure of the APTE_k plots gets similar to the others in the second half of the video. The other methods produce smoother estimates, with higher signal-to-noise ratios. This invalidates the APTE results for DROID-SLAM and validates the robustness of the Drunkard's Odometry pipeline to address the domain gap. The qualitative evaluation of visualizing the trajectories and the quantitative by the APTE complement each other. The former helps to realize when a method is producing totally wrong camera poses, and the latter to be able to objectively discretize which method is better when the trajectories look plausible. These comments have been added to the section of validation in real endoscopies.
> >
> > 6) The overview figure is quite complicated and less intuitive. I hope it includes some intermediate outputs as figure, rather than many text boxes.
> >
> > This is a good point and we agree that the original overview figure was too complex for a first or quick reading. Therefore, to increase the understandability of the general idea of our method, we added a simpler overview figure in the introduction section. We also decided to keep the previous overview figure in the method’s section, since we think a reader who wants to fully understand the details will appreciate this diagram that accompanies the mathematical formulations.

---

> > > ### Author Response · Authors · 2023-08-16
> > > **Part 3**
> > >
> > > 7) The colors in tables to highlight the results are a little confusing. I think using red / blue for first/second may be enough.
> > >
> > > We took into consideration this comment and tried different highlight versions. We find the color red is often linked to bad rather than good values. The applied gradual color gradient from green to yellow allows to highlight the order of more than two values intuitively and visually easy to interpret. In addition, the chosen color palette is more in concordance with the artistic style of the overview figures of the paper.
> > >
> > > [a] Sajjadi, M. S., Meyer, H., Pot, E., Bergmann, U., Greff, K., Radwan, N., ... & Tagliasacchi, A. (2022). Scene representation transformer: Geometry-free novel view synthesis through set-latent scene representations. In Proceedings of the IEEE/CVF Conference on Computer Vision and Pattern Recognition (pp. 6229-6238).

---

> > > > ### Comment · Reviewer_etHK · 2023-08-19
> > > > **My concerns are resolved and I'm raising my score.**
> > > >
> > > > Thank you for thoroughly dealing with all my concerns. I'm now much more convined about the practicality and motivation of this paper. This dataset is a good **middle ground** between practicality and domain gap. This dataset can be useful not only for endoscopies, but also other computer vision problems with non-rigid deformation aspects, as the authors answered; "isolating non-rigid aspects and having ground truth makes it easier to address this research challenge and makes our data also valuable for other tasks".
> > > >
> > > > And I'm happy that visualizing the trajectory can complement the APTE metric to evaluate the task without groundtruth. I believe this makes the authors' claim become stronger. One additional suggestion is that showing the original endoscopy video and **synced** trajectory reconstruction in supplementary video will be informative as many SLAM papers show camera trajectory and scene reconstruction with input video.
> > > >
> > > > Added infomormation about scene deformation and overview figure are also informative.
> > > >
> > > > Since my main concerns are resolved, I'm changing my rating to **"6: Marginally above acceptance threshold"**.
> > > > I appreciate the authors' effort.
> > > >
> > > > (minor typo: there seems no matching parenthesis on line 64. I believe the authors may thoroughly revise this kind of minor typos. but I'm notifying it since it sticked out to my eyes.)

---

> > > > > ### Author Response · Authors · 2023-08-25
> > > > >
> > > > > We really appreciate the reviewer's thoughtful and detailed revision of our paper. The feedback has been very valuable in refining our work, and we are pleased that the reviewer's concerns have been effectively addressed. The recognition of the practicality and motivation  of our paper, along with the subsequent rating improvement, is encouraging to us.
> > > > >
> > > > > We agree that the integration between the APTE metric and the trajectory visualization enhances the robustness of our proposed approach and reinforces our claims. We also thank the reviewer for the suggestion regarding a supplementary video showing the original endoscopy videos next to the synchronized trajectory reconstruction. We view it as a nice enhancement to our qualitative evaluation. As such, we have incorporated it as a novel animation into the figure depicting the estimated trajectories of the endoscopic videos. Additionally, we have included the corresponding video in the supplementary material.
> > > > >
> > > > > We apologize for the presence of minor typographical errors. We keep thoroughly revising our manuscript to eliminate such errors.
> > > > >
> > > > > To make it easier to see all the relevant changes made since the initial version, we have changed the text color of those parts to blue.

---

### Author Response · Authors · 2023-08-16

We would like to thank the reviewers for their efforts on handling our submission and providing such excellent feedback on our work. All comments have been very valuable to improve the quality of our manuscript and our work. We have addressed all comments and have made a thorough revision of our submission based on them. As part of our revision, we also run additional experiments that we will include in the final version of the paper if accepted. We are also grateful for that, as it was the reviewer’s feedback that motivated extra experiments. Please, find in the following our item-by-item responses under each author review, and notice the associated changes in the manuscript.

---

> ### Comment · Area_Chair_kV8y · 2023-08-29
> **Reviewers: please read authors' responses and share your thoughts and additional questions**
>
> Dear reviewers,
>
> While this paper got pretty consistent ratings, we still kindly request that you review the authors' rebuttal and indicate whether your concerns have been satisfactorily addressed.
>
> Thank you for your valuable contributions to this process.

---

### Decision · Program_Chairs · 2023-09-22

**Decision:**

Accept (Poster)

**Comment:**

This paper got positive ratings from all reviewers. A common concern among the reviewers is about the motivation and practical application of the dataset, specifically regarding camera pose estimation in deforming scenes. While endoscopies in medical applications could serve as a direct application, it is not clear to come up with other compelling applications.

In their rebuttal, the authors address these concerns, which successfully convince the reviewers. Apart from these concerns, the paper is in good shape overall. It includes a large-scale dataset and demonstrates its advantages through various experiments, including real endoscopies.

The AC concludes that the paper meets the NeurIPS D&B bar. The authors are advised to carefully review the feedback provided by the reviewers and enhance the paper's quality for the camera-ready version.